# The Impact of Adolescent Internet Addiction on Sexual Online Victimization: The Mediating Effects of Sexting and Body Self-Esteem

**DOI:** 10.3390/ijerph18084226

**Published:** 2021-04-16

**Authors:** Alicia Tamarit, Konstanze Schoeps, Montserrat Peris-Hernández, Inmaculada Montoya-Castilla

**Affiliations:** 1Department of Personality, Assessment and Psychological Treatment, Faculty of Psychology, University of Valencia, 46022 Valencia, Spain; alicia.tamarit@uv.es (A.T.); konstanze.schoeps@uv.es (K.S.); inmaculada.montoya@uv.es (I.M.-C.); 2Department of Personality, Evaluation and Psychological Treatments of the University of the Basque Country, 48940 Leioa, Spain

**Keywords:** internet addiction, body self-esteem, sexting, sextortion, grooming, structural equation modeling (SEM)

## Abstract

Adolescents’ problematic use of the internet and the risk of sexual online victimization are an increasing concern among families, researchers, professionals and society. This study aimed to analyze the interplay between adolescents’ addiction to social networks and internet, body self-esteem and sexual–erotic risk behavior online: sexting, sextortion and grooming. While sexting refers to the voluntary engagement in texting sexual–erotic messages, sextortion and grooming are means of sexual–erotic victimization through the use of the internet. Participants were 1763 adolescents (51% girls), aged 12 to 16 years (*M* = 14.56; *SD* = 1.16), from public (*n* = 1068; 60.60%) and private (*n* = 695; 39.40%) high schools in the Basque Country (Spain). We carried out structural equation modeling (SEM) using Mplus to assess the mediating effects of body self-esteem in the relationship between addiction to social media and internet and sexual–erotic risk behavior. The results showed that internet addiction predicts online sexual victimization; specifically, the best predictors of sexting, sextortion and grooming victimization were symptoms of internet addiction and geek behavior. Body self-esteem and sexting mediated the relationship between internet addiction and sexual online victimization in adolescents. These results highlight the importance of attending to adolescents’ mental health regarding their online behavior, considering the risk and protective factors involved, due to its close association with online sexual victimization.

## 1. Introduction

Internet and social media use has grown intensely in recent times, which has facilitated the development of a unique way of communication through information and communications technology (ICT) [1]. Social interactions online evolve into a different paradigm, where ICT users enjoy its many advantages while encountering new threats, unknown to analogic relationships [2]. There have been unprecedented problems, such as internet addiction or online victimization, which emerged as a result of the problematic use of internet as a novel medium for social interaction. Research has identified and thoroughly studied the impact of problematic internet use on psychosocial health and well-being in recent decades [3].

Young people are especially at risk of sexual online behavior or online victimization in the internet era [1]. Adolescence is a period of constant changes, on a cognitive, social, emotional and physical level, where social interactions with peers are one of the main sources of recognition and development of their identities. Internet and social media have provided powerful communication techniques, such as networking platforms, the possibility of using online profiles and long-distance audiovisual communication, which have become the core of adolescent’s lives and play an essential role in their psychological and social development [4].

With regard to this new paradigm, one of the most relevant issues in the study of adolescent well-being in recent decades has been their relationship with online technologies—adolescents who have grown up in the era of social media and technological advances have challenged the way psychology understands their behavior [5]. The use of the internet and social media has been a research area of increasing interest regarding youths’ emotional and social development, due to the intensive use of ICTs and omnipresence in adolescents’ interactions with their environment [6]. The internet and social networks are not only tools of communication and information exchange, but the way adolescents interact with each other over the internet has changed the primary risks to their psychological well-being, unprecedented before the digital era [7]. This is especially relevant for adolescents who are going through the early stages of puberty, which is typically between the ages of 12 and 16 years [8].

### 1.1. Internet Addiction and Problematic Social Media Use

Among these risks, research has focused on internet and social networking addiction, which is remarkably prevalent in this age group. Internet addiction in adolescents between 15 and 19 years of age has a prevalence of 4.39%, which in Spain is equivalent to more than 94,000 adolescents with problematic online behaviors [9].

Peris et al. [10] have identified four dimensions that comprise this kind of behavioral addiction: addiction symptoms, social media use, geek behavior and nomophobia. Internet addiction symptoms are characterized by the adolescent’s need or impulse for continuing to use the internet despite their desire to stop. This behavior is common in adolescents under 16 years, due to the highly addictive nature of social media, which shares psychopathologic signs with other types of addiction [11]. Social media use is related to the user’s interaction with others, generally peers, through online platforms, thus diminishing the frequency of face-to-face communication [10]. Another type of problematic internet use is geek behavior, which refers to conducts associated with an intense interest towards new technologies, specifically the access to information, online communication, social networking and more, resulting in maladaptive, problematic behavior [12]. It has been associated with gambling and gaming-related issues, usage of erotic websites and compulsive use of online communities, which is associated with psychological problems in minors [1,13]. Nomophobia is a concept that has also been addressed as “fear of missing out” (FOMO), which refers to the non-clinical fear of staying disconnected from the internet and not participating in online social interactions [14]. This phenomenon is associated with emotional distress and has been identified as a risk factor in adolescence, since it is another sign of internet addiction [12,15].

### 1.2. Sexual Online Victimization: Sexting, Sextortion and Grooming

The term sexting —a combination of “*sex*” and “*texting*”—is used in scientific literature as in everyday language and refers to the dissemination and expression of sexual content through digital media, for example, sending erotic pictures, engaging in sexual conversations or sharing audiovisual material through the internet [16]. Thus, it is associated with a problematic use of the internet and social media, functioning as a risk factor and posing a threat to adolescents’ well-being [3,17]. There are two dimensions of this construct: erotic sexting, which refers to content that is of a sensual, provocative, insinuating nature; and pornographic sexting, which includes explicit, often nude, sexual content [18]. This phenomenon, in recent years, has evolved as new technologies advanced, becoming highly prevalent in adolescents [19]. According to Rey et al. (2019), the prevalence among adolescents can vary highly depending on the study—between 4.6% and 31% of adolescents have sent sexual content, and these numbers increase with age [20]. Sexting can play a protective and risk role depending on the way adolescents engage in it [21]. Sending texts may be seen as a method of sexual expression, while receiving texts, especially when unsolicited or unwanted, may be seen as harassment [17,22]. This method of online communication is associated with a problematic use of the internet among adolescents [3]. While sexting can be considered a normative form of sexual communication, it has been linked to internet addiction signs, namely addiction symptoms, social media use and online gambling (a characteristic of problematic geek behavior), threatening mental health through maladaptive online behavior [23,24]. Research shows that adolescents undergo a particularly vulnerable period, where they are more likely to engage in risky behaviors to ensure their social adaptation—this kind of coping could lead them to use the internet in problematic ways, facilitating internet addiction and a higher prevalence of sexting among peers [25,26]

Sextortion is a concept derived from the study of erotic risk in online behavior and refers to cyber-harassment of sexual nature [27]. It refers to a specific kind of blackmail, using explicit sexual content as a means of manipulating the victim or forcing them to act in a way that would benefit the perpetrator [22]. It is strongly associated with the problematic use of the internet and social networks, and it predicts mental health problems such as depression, sexual risk behavior and self-esteem problems, thus significantly decreasing adolescents’ well-being [28]. It is different to sexting in the sense that sexting refers to sharing sexual content, while sextortion includes coercion and humiliation through sexting behavior [3]. Among the types of sexual victimization, research has identified erotic sextortion and coercive sextortion. The first type refers to experiencing erotic harassment through digital media, that is, the non-consensual dissemination of sexual content without the person’s permission; while the second includes an intentional manipulation, in other words, humiliation, violence or blackmail, by means of using one’s own sexual material online [27]. In Spain, the risk of online sexual victimization in adolescents aged 12 to 17 years ranges from 24% to 39%, according to different studies [28].

Traditionally, grooming has been studied as a process intended for sexual abuse and aggression, and it is a type of sexual victimization that can be applied to online interactions [29]. Adult cyber-groomers “prepare” their victims, generally minors, by befriending them, earning their trust as a way of seducing them and luring them into an intimate space that facilitates sexual abuse and increases the child or adolescent’s vulnerability [27,30]. It is associated with the concepts described above, since online sexual abuse includes sexting as a tool of sextortion, and grooming is another strategy that serves the aggressor in their perpetration of sexual violence [18]. In addition, it is directly related to problematic internet use and poses a threat to the victim’s well-being and mental health [3]. The risk of being a victim of online grooming is highly prevalent in Spanish adolescents, with approximately 12% reporting to have suffered from it [3].

Internet addiction entails the extensive use of online websites and social media, which becomes problematic for adolescents since it interferes with their everyday life, entailing serious risks for their well-being and social relationships [31]. Addiction symptoms, social media use and geek behavior are associated with risk factors for adolescents, such as substance consumption, visiting pornography and gambling-related sites or experiencing online victimization, threatening their physical and mental health [12,15]. These factors are also related to nomophobia; the extreme attention dedicated to the use of technological devices increases adolescents’ vulnerability in online settings [32]. Sexting has also been associated with online victimization in Spanish adolescents—sending and sharing sexual content through the internet has been observed to increase the likelihood of cyber-victimization, which can include experiencing sextortion and grooming [20]. Sexting, being a highly prevalent phenomenon [19,20], has become a priority in research on online behavior in adolescents, since it has been identified as a major high risk in problematic internet use and is highly correlated to suffering online sexual harassment and other forms of victimization [18].

### 1.3. The Role of Body Self-Esteem

Self-esteem, in this context, is manifested as one’s evaluation of their own worth through the use of internet and social media, thus, the assessment of a person’s concept of their own body is studied through the construct of body self-esteem [33]. Literature on this construct indicates that low self-esteem is associated with body self-esteem problems, which are emphasized by the increasing display of adolescents’ physical images on social media platforms [34]. Body satisfaction refers to a person’s positive perception of their own body, and it is negatively related to young people’s use of social networking sites, due to the high exposure of one’s virtual image online [35]. Physical attractiveness is derived from the concept of body self-esteem, and means to what extent one perceives their own body as attractive or aesthetically pleasing [36]. The impact of body self-esteem on adolescents’ well-being is relative, because adolescents can rate very specific areas of their body as attractive, while showing dissatisfaction with others—this has been observed to function as a protective mechanism towards external judgement [37,38]. Focusing on the areas they like about their bodies, rather than their general attractiveness, they find solace in the context of overt criticism on the internet [12]. Nonetheless, perceived physical attractiveness could be positively related to online vulnerability, since adolescents that rely on social reinforcement of their perceived attractiveness are more likely to increase the exposure of their online image, therefore being more accessible to online sexual offenders [39].

Moreover, low body self-esteem has been identified as a risk factor in online contexts; adolescents’ need for validation through the internet to compensate their low body satisfaction can lead them to engage in problematic behaviors, increasing their vulnerability to online sexual victimization [39,40]. However, further research is needed to study the nature of the relationship of body self-esteem to online problematic behavior and vulnerability. Self-esteem is often considered as a mediator in the study of adolescents’ well-being [8], and it has been observed to mediate the relationship between body mass index and internet addiction [41]. Furthermore, internet addiction has been identified as a significant predictor of disordered eating, due to its association with body image problems [42], and negative feelings caused by internet addiction are predictive of lower overall self-esteem [43]. Nonetheless, further research is necessary to understand the role of body self-esteem in adolescence and its relationship with internet problematic behavior. Thus, it would be of interest to study whether body self-esteem has a mediating effect on the interplay between internet addiction signs and online victimization, due to its crucial role in mental health and well-being in adolescence.

### 1.4. The Current Study

However, given the potential negative consequences and implication of online sexual victimization for adolescents’ well-being, there is currently no published research to date on its relationship with internet addiction and problematic use of social media. In addition, body self-esteem may have a crucial effect on the relationship between the risks associated with the use of the internet and well-being, since it is associated with online sexual victimization [18], therefore, it would be of interest in this study to analyze if body self-esteem could function as an independent variable or a mediator in a model that comprises the aforementioned variables. Path analysis on the interplay among these variables could reveal the underlying mechanism that may explain the relationship between problematic internet use and online victimization, highlighting the importance of their effect on adolescents’ well-being [3].

The purpose of this study was to analyze the association between internet addiction and sexual victimization in adolescents, and whether body self-esteem mediates this relationship. Drawing from the aforementioned research, we hypothesize that (1) there will be a direct positive effect of internet addiction on online victimization, namely erotic sextortion, coercive sextortion and grooming, (2) body self-esteem (including body satisfaction and physical attractiveness) and erotic and pornographic sexting will be associated with internet addiction, (3) these variables will explain some of the variance of sextortion and grooming, (4) sexting will have a mediating effect in the relationship between internet addiction and online victimization and (5) body self-esteem will play a significant role as an independent variable or a mediator in this mediational model.

## 2. Materials and Methods 

### 2.1. Participants

The participants were 1763 adolescents, selected using a simple random sampling technique from the list of education centers in the Basque Country. They were equally distributed between genders (50.99% girls), and their age ranged from 12 to 16 years (*M* = 14.56, *SD* = 1.16). The participants were studying in public (*n* = 1.068; 60.60%) and private (*n* = 695; 39.40%) high schools in the Basque Country (Spain), attending the following years of compulsory secondary education (E.S.O.): 20.76% of participants attended 2nd year of E.S.O.; 24.05% attended 3rd year of E.S.O.; 29.84% attended 4th of E.S.O.; and 25.35% were attending the 1st of year post-secondary education. In terms of ethnicity, the composition was as follows: 96.31% Basque Country, 0.91% other Spanish regions, 0.51% Europe, 1.93% South America, 0.34% China/Africa.

### 2.2. Variables and Instruments

#### 2.2.1. Socio-Demographic Variables

Personal and socio-demographic data regarding the students’ gender, age and high school were collected using ad hoc questions.

#### 2.2.2. Addiction to Social Networks and the Internet

Internet and online social networking addiction was measured using the scale of risk of addiction to social media and the internet for adolescents (ERA-RSI) [10]. The measurement consists of 29 items divided into four factors: symptoms of addiction, social media use, geek behavior and nomophobia. The first subscale, symptoms of internet addiction (9 items), measures the manifestations of addiction to non-toxic substances (e.g., “*I would feel angry right now if I had to do without using the internet*” or “*I have been losing sleep over social media and watching online shows*”). The second subscale, social media use (8 items), assesses common behaviors of adolescent “virtual socialization” (e.g., “*I use the chat*” or “*I check my friends’ profiles*”). The geek behavior dimension (6 items) refers to particular behaviors such as socializing in groups with specific interests, the practice of virtual and role-playing games, engaging in casual sexual encounters, etc. (e.g., or “*I spend time on social media and the Internet to play online and/or role-playing games*”). The nomophobia subscale (6 items) refers to anxious and controlling feelings when using a smartphone (e.g., “*If my messages are not answered immediately I feel anxious and distressed*” or “*I have a smartphone and I start feeling anxious or distressed when people don’t answer immediately to my messages*”). Scores range from 1: never or hardly ever; 2: sometimes; 3: quite often; and 4: often or always. The score for each dimension is obtained by adding the corresponding items and dividing by the number of items and doing the same for total addiction. The reliability analysis showed adequate indexes in this sample for the internet addiction (Table 1) symptoms scale (α = 0.83; average variance extracted (AVE) = 0.36; composite reliability coefficient (CRC) = 0.92) social media use (α = 0.69; AVE = 50; CRC = 94), geek behavior (α = 0.69; AVE = 0.28; CRC = 0.79), and nomophobia (α = 0.80; AVE = 0.36; CRC = 0.85).

#### 2.2.3. Body Self-Esteem

This scale has as its predecessor a preliminary scale composed of twelve items, nine of them grouped into three body zones (face, upper torso and lower torso), and another three belonging to general aspects, with which an overall score of body self-esteem was obtained [44]. The current body self-esteem scale (Escala de Autoestima Corporal, EAC) has been redesigned and two items have been added in each body area and in the general aspects, now called anthropometric items. In addition, six new items are proposed with an emotional evaluative content called physical attractiveness (PA). Thus, the EAC is now composed of 26 items with a unidimensional structure, but which assess body self-esteem in two aspects, body satisfaction (BS), the cognitive aspect (20 items), and physical attractiveness (PA), the emotional aspect (6 items). The first dimension, body satisfaction, is divided into four body areas: face (e.g., *How satisfied are you with your eyes/mouth?*), upper torso (e.g., “*How satisfied are you with your breasts/pectorals?*”), lower torso (e.g., “*How satisfied are you with your butt?*”) and anthropometry (e.g., *How satisfied are you with your height and size?*). It is scored through a 10-point Likert scale where 1 is “very dissatisfied” and 10 is “very satisfied”. The emotional dimension of body self-esteem, physical attractiveness (PA), is formed by 6 items and six aspects are evaluated: physically interesting, socially charming, sexy, attractive, sensual and erotic (e.g., “*To what extent do you consider yourself a physically attractive person?*”), on a 10-point Likert scale where 1 is “not attractive at all” and 10, “very attractive”. This scale shows adequate psychometric properties in the studied sample (body satisfaction α = 0.93; AVE = 0.41; CRC = 0.93; physical attractiveness: α = 0.92 AVE = 0.67; CRC = 0.92).

#### 2.2.4. Sexual Online Victimization

*Sexting*. The Sexting Scale [28] is grouped into two dimensions: erotic sexting, which consists in sending and posting seductive and erotic photographs on social media (9 items, e.g., “I choose pictures to publish on social media where I look sexy”), and pornographic sexting, which consists of sharing nude or semi-nude photographs with their peers (4 items, e.g., “*My friends and I, we sent each other pictures in underwear*”). Both are direct and active forms of sexting, which has been observed to increase the risk of online sexual behavior. The items are scored on a four-point Likert scale ranging from 1 (never or nothing) to 4 (always or a lot) of how often they share sexual–erotic “pics” on social media or with their peers. This instrument, in its original version [28], has good psychometric properties, which is confirmed in the studied sample: erotic sexting (α = 0.89; AVE = 0.53; CRC = 0.81) and pornographic sexting (α = 0.79; AVE = 0.53; CRC = 0.81).

*Sextortion*. The Sextortion Scale [28] is composed of 10 items with a bifactor structure. The first factor, consisting of 5 items, is called erotic sextortion. The second factor, consisting of another 5 items, is called coercive sextortion. Erotic sextortion is headed by this statement: “*Sometimes messages, photos or videos are sent that can hurt. In your case …*” This statement is followed by 5 items, 3 of which specify whether one has been erotically harassed, teased or humiliated, and 2 items analyze whether one has observed it oneself. The second factor, coercive sextortion, is headed by the following sentence: “*Photographs with an erotic or sexual touch are published by …*” The statement is followed by a series of words, corresponding to the 5 items, in which the reasons for sextortion are proposed: blackmail, revenge, intimidation, etc. Thus, the erotic harassment items imply, directly or indirectly, suffering erotic harassment, as this factor has been named, by receiving, sending or uploading photographs, messages or other publications on the internet. The coercion dimension implies what the name of the scale suggests, that these erotic or sexual contents are uploaded with the intention to blackmail, intimidate or humiliate, for revenge or because others put on pressure to do so. The items are answered on a four-point Likert scale ranging from 1 (never or not at all) to 4 (always or a lot). The reliability of the scale was adequate for erotic sextortion (α = 0.70; AVE = 0.43; CRC = 0.76) and coercive sextortion (α = 0.73; AVE = 0.42; CRC = 0.76).

*Grooming*. The Grooming Scale [28], comprising 13 items, was administrated in order to assess the risk of online grooming victimization. The participants were given the following instructions “Think if ever someone quite older than you has tried to meet you and get closer to you, making you feel special, more valued, loved and even sending you gifts using internet, webcam, online chat, social media, etc.” They were then asked to estimate to which extent they recognize the following situation involving exhibitionist behaviors by their own choice or at the request of adults (e.g., “*I’ve shown my genitals or my naked body on the internet*”, “*I’ve been offered gifts or money in exchange for showing my naked body on the internet*”, “*An adult has tried to meet me with unclear erotic intentions*”). Each item was rated on a four-point Likert scale (1 = never or nothing, 4 = always or a lot). Peris and Maganto [10] provided satisfactory psychometric values. Adequate reliability indices were achieved in the sample of the present study (α = 0.86; AVE = 0.34; CRC = 0.87).

### 2.3. Procedure

The present study follows the ethical regulations of the Declaration of Helsinki [45], the guidelines of the Ethics Commission for Research on Human Beings of the University of the Basque Country/Euskal Herriko Unibersitatea (UPV/EHU) and the State Government. This research also follows the deontological guidelines of the Official College of Psychologists for experimentation on human beings. The study was approved by the Ethics Commission for Research with Human Beings (CEISH) of the University of the Basque Country/Euskal Herriko Unibersitatea (CEISH/136/2012/PERIS HERNANDEZ). 

Data were collected after requesting informed consent from school principals, parents and legal guardians and the participants themselves. Psychological examiners were trained in the information and application of the assessment instruments. The data collection took place in the schools during school hours and under similar conditions of information, motivation and time (approximately 40 min). 

### 2.4. Data Analysis

Descriptive analysis was carried out to analyze the means, standards deviations, asymmetry and kurtosis of the studied variables. Then, the relationship among all variables was studied through Pearson correlations. Structural equation modeling (SEM) analyses were conducted to analyze the mediating role of body self-esteem and sexting in the relationship between internet addiction symptoms and online sexual victimization. Goodness-of-fit was estimated using the following estimation indices: chi-square test, (χ^2^), comparative fit index (CFI), root mean square error of approximation (RMSEA) and standardized root mean square residuals (SRMRs). The mediational model was carried out with maximum likelihood estimation (MLR) and robust standard errors. Reliability indices were calculated for all variables, first with the assessment of Cronbach’s alpha, and then through a confirmatory factor analysis (CFA), from which the average variance extracted (AVE) and composite reliability coefficient (CRC) were calculated. For descriptive analysis and Pearson correlations, SPSS (version 26.0) was used. The software used for the SEM was MPlus (version 6.12) [46].

## 3. Results

### 3.1. Descriptive Statistics and Correlations between Variables

The results of descriptive analyses (Table 1) indicated adequate levels of asymmetry and kurtosis, within a ±2 range, which indicates normal distribution, except for pornographic sexting, erotic sextortion and grooming. 

Intercorrelations within dimensions (Table 2) were all significant and positive: all four subscales of internet addiction, body satisfaction and physical attraction, both variables of sexting and both variables of sextortion were positively and significantly correlated. All intercorrelations were moderate—the dimensions from the same questionnaire are associated with each other, but moderately, since they measure different constructs.

Addiction symptoms are significantly associated with the rest of the variables (indices ranging from *r* = 0.06 to *r* = 0.40), all with positive associations, except for body satisfaction, which is negative. Social media use was positively correlated with physical attractiveness, erotic sexting, erotic harassment and grooming and geek behavior (indices ranging from *r* = 0.11 to *r* = 0.29) and geek behavior was positively correlated with physical attractiveness, both variables of sexting, erotic sextortion and grooming (indices ranging from *r* = 0.15 to *r* = 0.29). Body satisfaction was positively associated with erotic sexting and negatively associated with both variables of sextortion, its association with sexting being moderate and with sextortion, weaker. Physical attractiveness was positively and moderately associated with erotic and pornographic sexting and grooming, showing a lower correlation with sextortion. As for sexting, sextortion and grooming, all variables were positively and significantly correlated (showing low to moderate indices ranging from 0.06 to 0.48), except for erotic sexting and coercive sextortion.

### 3.2. Structural Equation Modeling (SEM) with Mediation Analysis

In order to test the hypothesized association between internet addiction and sexual victimization in adolescents and whether body self-esteem functions as an independent variable or a mediator in this model, we performed two competitive SEMs with mediation analysis.

In a first step, a total mediation model was estimated, including the four dimensions of addiction to social networks and the internet (symptoms of addiction, social media use, geek behavior and nomophobia) and body self-esteem (body satisfaction and physical attraction) as predictor variables, sexual online victimization (erotic and coercive sextortion and grooming) as dependent variables and types of sexting (erotic and pornographic) as mediating variables. In this model, the covariances between predictor variables (internet addiction and body self-esteem) and dependent variables (sextortion and grooming victimization) were indicated. The first full mediation model showed unsatisfactory goodness-of-fit indices (*χ*^2^ (df) = 259.878 (18), CFI = 0.87; TLI = 0.70; RMSEA = 0.09, SRMR = 0.06). Consequently, a number of direct effects, which were proposed by Mplus as model modification indices, were included in the equation. This partial mediation model showed satisfactory indices of adjustment (*χ*^2^ (df) = 38.18 (14), CFI = 0.987; TLI = 0.96; RMSEA = 0.09, SRMR = 0.02) (Figure 1).

In a second step, a competitive mediation model was estimated, including the four dimensions of addiction to social networks and the internet (symptoms of addiction, social media use, geek behavior and nomophobia) as predictor variables, sexual online victimization (erotic and coercive sextortion and grooming) as dependent variables and types of sexting (erotic and pornographic) and body self-esteem (body satisfaction and physical attraction) as a mediating variable. This full mediation model showed acceptable goodness-of-fit indices (*χ*^2^ (df) = 60.49 (12), CFI = 0.94; TLI = 0.76; RMSEA = 0.0, SRMR = 0.05) (Figure 2).

However, modification indices suggested direct effects between internet addiction and online sexual victimization variables. Thus, a partial mediation model was tested (Figure 3), which showed excellent indices of model fit (*χ*^2^ (df) = 12.27 (8), CFI = 0.995; TLI = 0.97; RMSEA = 0.04, SRMR = 0.01). This model adjusted better to the data and was more parsimonious than the first one. Therefore, this is the model that was retained.

The association between body satisfaction and internet addiction was significant for addiction symptoms and geek behavior (β = −0.24 and β = 0.09) while, regarding physical attraction, it was only significant for geek behavior (β = −0.12). As for the sexting variables, erotic sexting was associated with addiction symptoms and geek behavior (β = −0.25 and β = 0.15), while pornographic sexting was related to addiction symptoms, social addiction and geek behavior (β = −0.25, β = −0.16 and β = 0.22). These results indicate that both body self-esteem and sexting are associated to some extent with internet addiction (Table 3). 

Erotic sextortion was associated with pornographic sexting and addiction symptoms (β = −0.23 and β = 0.19), coercive sextortion was only associated with body satisfaction (β = −0.17) and grooming was significantly related to all variables except for geek behavior and nomophobia (β *_EroticSexting_* = 0.21, β *_PornographicSexting_* = 0.29, β *_BodySatisfaction_* = −0.13, β *_PhysicalAttraction_* = 0.21, β *_SignsofAddiction_* = 0.18 and β *_SocialAddiction_* = 0.10).

The indirect effect of geek behavior on erotic sextortion was significantly mediated by pornographic sexting (β = 0.05, *p* = 0.02). Regarding grooming, the indirect effect of addiction symptoms on this variable was partially mediated by body satisfaction (β = 0.03, *p* = 0.047) and erotic sexting (β = 0.05, *p* = 0.003). Additionally, the indirect effect of geek behavior on grooming was partially mediated by erotic (β = 0.03, *p* = 0.02) and pornographic sexting (β = 0.07, *p* = 0.006). These results show that body satisfaction, erotic sexting and pornographic sexting play a mediating role in the relationship between internet addiction and online victimization.

The combination of direct and indirect effects explains 17.3% of the variance of erotic sextortion, 2% of the variance of coercive sextortion and 39.4% of the variance of grooming.

## 4. Discussion

Adolescents engage in social networking platforms and the internet on a daily basis as a regular communication and information tool, disregarding the negative consequences and potential risk for their mental health and well-being. Problematic internet use in youth has been a major concern among parents, teachers and professionals, yet the scientific evidence on this research topic is scarce. Evidently, more empirical studies are needed that shed light on the psychological mechanism underlying the association between addiction to the internet and social networks and the risk of online sexual victimization such as sexting, sextortion and online grooming. Thus, the present study aimed to examine the mediating role of sexting and body self-esteem in the relationship between internet addiction and sexual victimization in adolescents. The findings provide a more extensive and deeper understanding of the psychological risk factors of online victimization, which are especially relevant for healthy adolescent development. 

The results from structural equation modeling partially support the first hypothesis, indicating a statistically significant effect of internet addiction on online victimization. Specifically, addiction symptoms increase the risk of erotic sextortion and online grooming, while social media use has the same positive effect on grooming victimization only. In contrast, geek behavior and nomophobia have no direct effect on sexual online victimization according to the results in this adolescent sample. These findings are consistent with previous research, showing that adolescent internet addiction is associated with online victimization, such as cyber-bullying [15]. However, our results deviate from other studies that indicate that nomophobia and the related fear of missing out (FOMO) would increase the risk of experiencing potentially reputational or psychological harm as a result of online activities (online victimization) [32].

The results obtained in this study also partially support the second hypothesis, showing that body self-esteem and sexting are associated with internet addiction. On the one hand, our results revealed significant pathways between addiction symptoms and body satisfaction, indicating that adolescents’ addictive use of the internet and social media might be mitigating their perceived body satisfaction. On the other hand, geek behavior is positively associated with both dimensions of body self-esteem, that is, problematic internet behavior related to gambling and gaming is associated with a more positive body self-perception. Our findings are in line with previous studies, which suggest that extensive social engagement on social media sites negatively influences young people’s satisfaction with their bodies and appearance [40]. However, a recent study conducted by Veldhuis et al. [35] revealed that body image (but not body satisfaction) and social networking site use seem to mutually affect and reinforce one other in teenage girls. Thus, a positive body image not only is the result of intensive social media use, but also motivates individuals to engage in such social online behavior more often to promote acceptance of their own body, which might further empower and affirm them. In addition, Matthews et al. [38] found that male and female gamers improved their feelings about their bodies after playing a video game featuring hyper-idealized characters, which may prevent social comparison.

Regarding sexting behavior, our results indicate that addiction symptoms and geek behavior are significant predictors of erotic sexting, while those two in combination with social media use significantly predict pornographic sexting, too. The most relevant predictor seems to be addiction symptoms for both subtypes of sexting. Our findings corroborate previous studies that have shown that abusive internet use increases sexting among adolescents and young people, which is a potential risk behavior on the internet [23,26].

The third hypothesis has been supported by the results from SEM analyses, indicating that body self-esteem and sexting are significant predictors of online sexual victimization. Body satisfaction is negatively associated with coercive sextortion and grooming victimization, while physical attractiveness is positively related to grooming. Thus, adolescents who are satisfied with their body in general are less likely to be a victim of online harassment. However, adolescents who perceive themselves as physically attractive are at risk of being a victim of online grooming. These unexpected results are, however, consistent with certain existing research, which suggests that adolescents, particularly female teenagers, with body image concerns are at greater risk of online sexual victimization [39]. It is likely that teenagers who are dissatisfied with their body image use social media more often, adopting unsafe online behavior such as expanding their online network of social contacts indiscriminately and developing intimate relationships with strangers who might be sexual attackers. Furthermore, the results of a recent study conducted by Schoeps et al. [18] indicated that adolescents with a physically attractive body image are more likely to send erotic pictures over the internet, which makes them more vulnerable to capture by groomers. This might explain the positive association between physical attractiveness and increased risk of online victimization. 

Furthermore, our results show that both erotic and pornographic sexting are significant risk factors for online sexual victimization, including sextortion and grooming. These findings are in line with previous research, suggesting that adolescents who engage in sexting behavior voluntarily are more likely to fall into the trap of online groomers and sexual harassers [18]. Undoubtedly, sending and sharing erotic or even pornographic pictures online is a new trend among teenagers, ignoring the risk that those pictures might be used by sexual harassers or pornographic networks for multiple online victimization incidents.

The results of this study support the fourth hypothesis, confirming the mediating effect of sexting in the relationship between internet addiction and online victimization. Thus, our results indicate an indirect effect of geek behavior on erotic sextortion that was significantly mediated by pornographic sexting. In other words, adolescents who engage in problematic internet behavior such as gambling and gaming are more likely to send pornographic messages or pictures over the internet, which increases the risk of becoming a victim of erotic sextortion. In addition, the indirect effect of addiction symptoms on grooming victimization was partially mediated by erotic sexting. That is, those adolescents who use the internet impulsively, not being able to disconnect, also send erotic messages or pictures over the internet more often, which makes them an ideal target for online groomers. Moreover, the indirect effect of geek behavior on grooming was partially mediated by erotic and pornographic sexting. In other words, adolescents who engage in problematic internet behavior such as gambling and gaming are more likely to send erotic or even pornographic messages or pictures over the internet, which increases their risk of becoming a victim of online grooming. 

Finally, two competitive models were tested to determine the role of body self-esteem as an independent or mediating variable. Results from the SEM showed that the partial mediation model, including the four dimensions of addiction to social networks and the internet as predictor variables, sexual online victimization as dependent variables and sexting and body self-esteem as mediating variables, adjusted better to the data and was more parsimonious than the one that suggested body self-esteem as a predictor variable. In this model, the indirect effect of addiction symptoms on grooming victimization was partially mediated by body satisfaction. This means that adolescents with problematic internet use and feeling dissatisfied with their body may be searching for external approval in social networking platforms and chat rooms, which makes them an easy target for online sexual victimization. 

These results provide additional evidence to that found in previous studies, that have identified addiction and social networking, body dissatisfaction and sexting as potential risk factors of online sexual victimization separately [18,23,34,40,43]. Thus, our findings provide a comprehensive insight into the psychological mechanisms that may underlie the association between problematic use of the internet and sexual online victimization, including grooming and sextortion, which could have an important impact on adolescents’ well-being and mental health.

### 4.1. Strengths, Limitations and Future Research

This study presents several strengths, both theoretical and methodological. On the one hand, the study focuses on online sexual victimization considering different types of risks in the same multidimensional approach. Thus, sexting involves sending erotic pictures, engaging in sexual conversations or sharing audiovisual material through the internet, which might be considered a normative form of sexual expression, but has often been linked to unsafe online behavior in adolescents. A possible outcome of such online risk behavior is sextortion, which refers to cyber-harassment of an erotic or even coercive nature. Victims of sextortion often experience non-consensual dissemination or even blackmail by means of one’s own sexual material. In the case of grooming, sexting and sextortion are often used as tools in order to lure the minor into a trap of which the purpose is to seduce and then sexually abuse the victim. On the other hand, the methodological approach using structural equation modeling with mediation analysis allows us to test this multidimensional model, exploring the different pathways that might explain the psychological mechanism underneath online sexual victimization, all in one prediction model. 

Despite these strengths, the study is not without limitations. Regarding the sample, although representative for the Basque Country where the study was conducted, the results might not be generalizable for the whole of the Spanish territory. Future studies could replicate the findings on a national or even international level. Additionally, only self-reported data were used in this study and, although adolescents are considered to be reliable informants of their internal states and behaviors, including objective measures of either the dependent or independent variables would be beneficial for futures studies. This could reduce the common method bias, which refers to moderate associations between the variables studied, being attributed in part to the use of the same data collection method, in this case self-reports. With regard to the study design, a cross-sectional design was used, not allowing the determination of cause–effect relationships between the variables studied. Therefore, future research based on longitudinal data is warranted to establish causality relationships. Finally, the methodological approach used in this study is one possibility among many, and additional analyses of predictive models that can provide further explanations of online sexual victimization should be conducted in order to confirm the hypothetical model proposed by the authors.

### 4.2. Implications 

Although the present study is not without limitations and additional research is needed, the implications of this study are both theoretical and practical. Overall, the study makes an exceptional contribution to the literature by exploring multiple risk factors and pathways to sexual online victimization, which may allow for more successful intervention and prevention strategies. Continuing this line of thought, comprehensive programs for parents, teachers and health professionals should be able to unfold the mechanisms that underlie the risk of sexual online victimization so as to provide useful tools for their role as guides and overseers of adolescents’ sexual online behavior. Given that online victimization is difficult to uncover because of the anonymity of social media platforms, it is important to provide teenagers with social and emotional skills that foster their personal resources in order to protect them from negative consequences to their well-being and mental health.

The role of healthy images of adolescents’ bodies and physical appearances is crucial in light of our findings. Although social media and social networking sites are an important source of information and social interaction, they should not be the only one. Families and schools ought to work together, with the aim of providing real-world models that teach diversity and acceptance and tolerance of individual differences in internal and external features. Adolescents and even younger teenagers seek approval through “likes” of their online posts, which constitute a marker of peer status and popularity, as well as a sign of complying with models considered physically attractive. The work towards educational programs that focus on a positive use of technology and particularly social networking sites should start with accepting that they are an essential part of young people’s life and therefore integrate them in today’s professional and educational curriculum. 

Moreover, the study offers a more comprehensive conceptualization of internet and social networking addiction, considering different dimensions with different outcomes regarding sexual online victimization. Thus, the most relevant elements seem to be addiction symptoms associated with the need or impulse to use the internet and geek behavior, which is related to online gambling and video gaming. Both reflect adolescents’ most problematic online practices, when engaging in what starts as a leisure activity but easily becomes an obsessive habit with important negative outcomes for their mental health.

## 5. Conclusions

The findings of this study extend previous literature, providing a novel approach which analyzes addiction to the internet and social networking as a potential risk factor for online sexual victimization in adolescence, considering the mediating role of body self-esteem and sexting. Individuals who engage in addictive internet use increase their risk of sextortion and grooming victimization, while body dissatisfaction and frequent sexting practices aggravate the effect. Consequently, adolescents’ well-being and mental health are at risk when falling into the trap of online sexual harassers and groomers. Apparently, problematic internet use is common among adolescents, and so is online sexual harassment if parents, teachers and professionals miss the chance of preparing individuals to deal with such precarious situations, including asking for help and reporting the incident. Finally, this study has identified important risk factors that underlie the psychological mechanisms of sexual online victimization, which might help to reduce their impact on adolescent victims.

## Figures and Tables

**Figure 1 ijerph-18-04226-f001:**
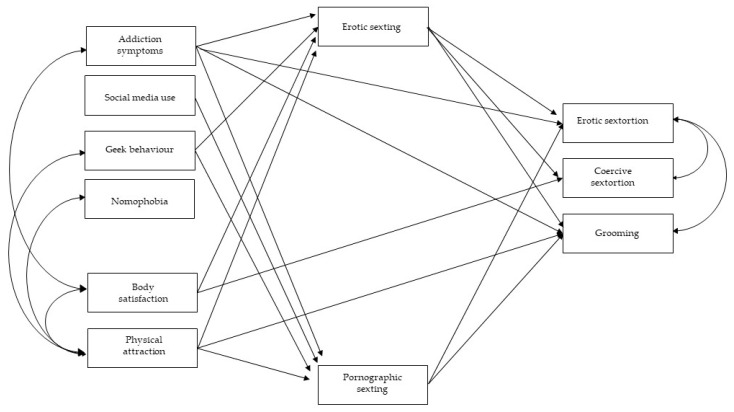
Tested structural equation model with partial mediation: interplay between internet and social networking addiction, body self-esteem and sexual online victimization mediated by sexting. *Note*. Continuous pathways are significant at *p* < 0.01. The relationships of the factors with their indicators have not been drawn for simplicity; non-significant pathways were not included.

**Figure 2 ijerph-18-04226-f002:**
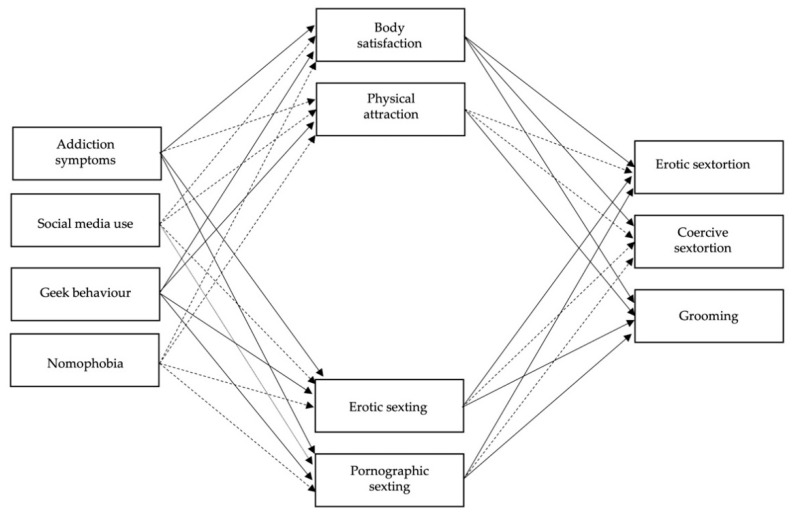
Tested structural equation model with complete mediation: interplay between internet and social networking addiction and sexual online victimization mediated by body self-esteem and sexting. *Note*. Continuous pathways are significant at *p* < 0.01, dotted pathways are not significant. The relationships of the factors with their indicators have not been drawn for simplicity.

**Figure 3 ijerph-18-04226-f003:**
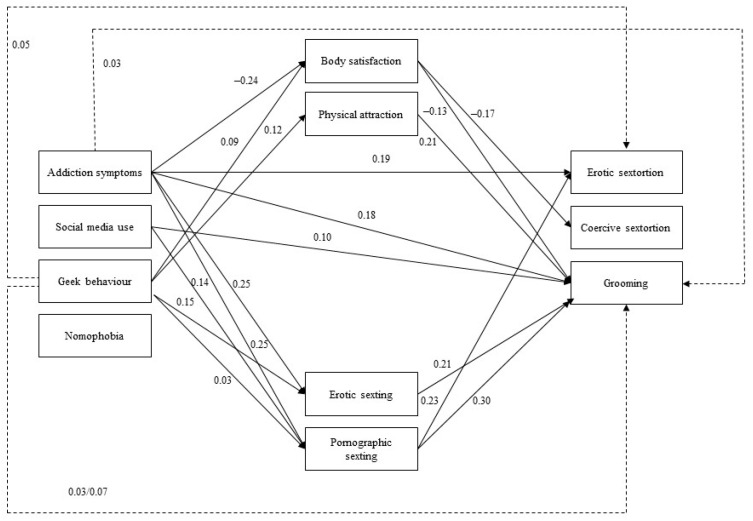
Tested structural equation model with partial mediation: interplay between internet and social networking addiction and sexual online victimization mediated by body self-esteem and sexting. *Note*. Significant effects shown as standardized coefficients (β); continuous pathways are significant at *p* < 0.01; dotted pathways indicate the indirect effects; non-significant pathways were not included.

**Table 1 ijerph-18-04226-t001:** Descriptive analyses of the studied variables.

	Range	Minimum	Maximum	Mean (SD)	Asymmetry	Kurtosis
Addiction symptoms	27	9	36	18.21 (5.60)	0.40	−0.39
Social media use	24	8	32	21.17 (4.86)	−0.09	−0.38
Geek behavior	13	6	19	9.31 (2.77)	1.05	0.77
Nomophobia	16	6	22	12.62 (4.20)	0.05	−0.93
Body satisfaction	8	2	10	6.79 (1.33)	−0.26	0.13
Physical attractiveness	9	1	10	6.19 (1.66)	−0.43	0.09
Erotic sexting	27	9	36	17.13 (5.44)	0.58	−0.07
Pornographic sexting	12	4	16	4.63 (1.54)	3.66	16.58
Erotic sextortion	15	5	20	6.62 (2.11)	2.08	5.96
Coercive sextortion	15	5	20	8.84 (3.08)	0.68	0.18
Grooming	38	13	51	17.20 (5.23)	2.28	6.91

**Table 2 ijerph-18-04226-t002:** Reliability coefficients and correlations between variables.

	1	2	3	4	5	6	7	8	9	10	11
1. AS	–										
2. SMU	0.56 **	–									
3. GB	0.29 **	0.30 **	–								
4. No	0.61 **	0.55 **	0.32 **	–							
5. BS	−0.19 **	−0.02	0.06	−0.03	–						
6. PA	0.06 **	0.11 *	0.15 **	0.13 **	0.61 **	–					
7. ES	0.30 **	0.21 **	0.23 **	0.23 **	0.18 **	0.45 **	–				
8. PS	0.19 **	0.05	0.25 **	0.13 **	0.03	0.18 **	0.34 **	–			
9. EST	0.30 **	0.21 **	0.15 **	0.15 **	−0.08 **	0.09 **	0.23 **	0.22 **	–		
10. CST	0.09 **	0.04	0.07	0.06	−0.09 **	−0.02	0.04	0.06 **	0.24 **	–	
11. Gr	0.40 **	0.29 **	0.29 **	0.25 **	0.01	0.28 **	0.43 **	0.42 **	0.48 **	0.10 **	–
*α*	0.82	0.83	0.69	0.80	0.93	0.92	0.89	0.79	0.70	0.73	0.86

*Note.* AS = Addiction symptoms. SMU = Social media use. GB = Geek behavior. No = Nomophobia. BS = Body satisfaction. PA = Physical attractiveness. ES = Erotic sexting. PS = Pornographic sexting. EST = Erotic sextortion. CST = Coercive sextortion. Gr = Grooming. * *p* < 0.05. ** *p* < 0.01.

**Table 3 ijerph-18-04226-t003:** Coefficients of total, direct and indirect effects.

DV	IV	M	Total Effect			Total Indirect Effect			Specific Indirect Effect		
			c	SE	CI 95%	c’	SE	CI 95%	ab	SE	CI 95%
Erotic Sextortion	Addiction Symptoms	Body satisfaction	0.28 ***	0.07		0.09 *	0.04		-	-	-
		Physical attractiveness							-	-	-
		Erotic sexting							-	-	-
		Pornograph. sexting							-	-	-
Erotic Sextortion	Social Media Use	Body satisfaction	0.03	0.06		−0.04	0.02		-	-	-
		Physical attractiveness							-	-	-
		Erotic sexting							-	-	-
		Pornograph. sexting							-	-	-
Erotic Sextortion	Geek Behavior	Body satisfaction	0.06 *	0.03	0.01, 0.12	0.06 *	0.03	0.01, 0.012	-	-	-
		Physical attractiveness							-	-	-
		Erotic sexting							-	-	-
		Pornograph. sexting							0.05 *	0.03	0.002, 0.10
Erotic Sextortion	Nomophobia	Body satisfaction	0.003	0.02		0.003	0.02		-	-	-
		Physical attractiveness							-	-	-
		Erotic sexting							-	-	-
		Pornograph. sexting							-	-	-
Coercive Sextortion	Addiction Symptoms	Body satisfaction	0.05	0.03		0.05	0.03		-	-	-
		Physical attractiveness							-	-	-
		Erotic sexting							-	-	-
		Pornograph. sexting							-	-	-
Coercive Sextortion	Social Media Use	Body satisfaction	−0.01	0.02		−0.01	0.02		-	-	-
		Physical attractiveness							-	-	-
		Erotic sexting							-	-	-
		Pornograph. sexting							-	-	-
Coercive Sextortion	Geek Behavior	Body satisfaction	−0.002	0.02		−0.01	0.02		-	-	-
		Physical attractiveness							-	-	-
		Erotic sexting							-	-	-
		Pornograph. sexting							-	-	-
Coercive Sextortion	Nomophobia	Body satisfaction	−0.004	0.01		−0.004	0.01		-	-	-
		Physical attractiveness							-	-	-
		Erotic sexting							-	-	-
		Pornograph. sexting							-	-	-
Grooming	Addiction Symptoms	Body satisfaction	0.35 ***	0.06	0.08, 0.23	0.17 ***	0.04	0.24, 0.44	0.03 *	0.02	0, 0.06
		Physical attractiveness							-	-	-
		Erotic sexting							0.05 **	0.02	0.02, 0.08
		Pornograph. sexting							-	-	-
Grooming	Social Media Use	Body satisfaction	0.05	0.06	0.24, 0.44	−0.05	0.03	−0.10, −0.004			
		Physical attractiveness							-	-	-
		Erotic sexting							-	-	-
		Pornograph. sexting							-	-	-
Grooming	Geek Behavior	Body satisfaction	0.11 **	0.03	0.04, 0.16	0.11 **	0.03	0.04, 0.16	-	-	-
		Physical attractiveness							-	-	-
		Erotic sexting							0.03 *	0.01	0.01, 0.05
		Pornograph. sexting							0.07 **	0.02	0.02, 0.10
Grooming	Nomophobia	Body satisfaction	0.01	0.03	−0.05, 0.06	0.01	0.03	−0.05, 0.06	-	-	-
		Physical attractiveness							-	-	-
		Erotic sexting							-	-	-
		Pornograph. sexting							-	-	-

*Note.* * *p* < 0.05. ** *p* < 0.01 *** *p* < 0.001.

## Data Availability

There is no data available for this study.

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
