# Peer review of "The Impact of Adolescent Internet Addiction on Sexual Online Victimization: The Mediating Effects of Sexting and Body Self-Esteem"

_ijerph, 2021, doi:10.3390/ijerph18084226_

Round 1
Reviewer 1 Report
This is a good study which provides new insight regarding the associations between social networking and internet addiction and the role of body self-esteem. The paper is well written, clear and readable and suitable for the readers of the journal. However, I suggest some clarifications and elaborations before it fully meets the standards required for publication (see below).
Comments specific to the different parts of the paper:
Introduction:
The literature review seems appropriate for the topic. The paper does engage with current literature in the area. However, I suggest the authors: specify and make it clear to which period of time they are referring to when they talk about adolescence: 12-16? The literature section of the paper is written using general terms such as: this age, this age group... but the ages are not mentioned until the reader reaches the method section of the paper.
Also, I suggest authors complement the theoretical information presented with different studies prevalence rates.
Method:
The methodology used is appropriate for addressing the research hypothesis and is clearly explained in the article. Some questions remain unanswered, however:
As participation was voluntarily, what was the attrition rate of children at the participating schools?
How (orally, written or both), where and by whom were the adolescents themselves informed about the study and how did they give their consent to participating?
Did schools receive detailed information regarding the results obtained in the study?
Reviewer 2 Report
The interesting manuscript described the associations among social media and internet addiction, body self-esteem, sexting, sextortion, and grooming in adolescents. However, there are several questions should be taken into consideration.
- I agree with the close associations between the characteristics mentioned above, however, it should be cautious to state the mediating effects of the sexting and body self-esteem based on a cross-sectional study. As being reported in the previous studies (including which was cited in the manuscript , references 34), body self-esteem might mutually affect internet addiction, well-being, and online problematic behaviors, we would also suppose that the body self-esteem "predict" internet addiction and was predicted by the online sextortion etc. Such a hypothesis would also be tested and supported by the data collected in the present cross-sectional study.
- Common method bias could not be excluded, because all of the data were relied on participants' self-reports. That is, the moderate associations between different variables were- partially attributed to the same method of data collection, i.e., self report.
- Could authors check the results in the Figure 2? Two associations between addiction symptoms and grooming were listed in the figure. Both of the two lines seem to indicate same direction.
Round 2
Reviewer 2 Report
1.The comment 1 was not fully addressed, The argument in the introduction is unpersuasive. Authors did not list adequate evidence to support the mediating hypothesis. Perhaps, another competitive model, including moderate effect model, would be supposed and tested.
2.The comment 2 was not well addressed. What the so-called common method bias concerned is that both of the dependent and independent variables were evaluated by same method. One potential method to diminish such a bias is that dependent variable based on objective assessment and the independent variables based on self-reported data, and vice versa.
